# Overcoming Challenges for CD3-Bispecific Antibody Therapy in Solid Tumors

**DOI:** 10.3390/cancers13020287

**Published:** 2021-01-14

**Authors:** Jim Middelburg, Kristel Kemper, Patrick Engelberts, Aran F. Labrijn, Janine Schuurman, Thorbald van Hall

**Affiliations:** 1Department of Medical Oncology, Oncode Institute, Leiden University Medical Center, 2333 ZA Leiden, The Netherlands; j.middelburg@lumc.nl; 2Genmab, 3584 CT Utrecht, The Netherlands; kke@genmab.com (K.K.); pen@genmab.com (P.E.); ala@genmab.com (A.F.L.); jsc@genmab.com (J.S.)

**Keywords:** antibody therapy, immuno-oncology, CD3-bispecific antibody, T-cell engager, solid tumors, on-target off-tumor toxicity, T-cell co-stimulation, tumor-associated antigens

## Abstract

**Simple Summary:**

CD3-bispecific antibody therapy is a form of immunotherapy that enables soldier cells of the immune system to recognize and kill tumor cells. This type of therapy is currently successfully used in the clinic to treat tumors in the blood and is under investigation for tumors in our organs. The treatment of these solid tumors faces more pronounced hurdles, which affect the safety and efficacy of CD3-bispecific antibody therapy. In this review, we provide a brief status update of this field and identify intrinsic hurdles for solid cancers. Furthermore, we describe potential solutions and combinatorial approaches to overcome these challenges in order to generate safer and more effective therapies.

**Abstract:**

Immunotherapy of cancer with CD3-bispecific antibodies is an approved therapeutic option for some hematological malignancies and is under clinical investigation for solid cancers. However, the treatment of solid tumors faces more pronounced hurdles, such as increased on-target off-tumor toxicities, sparse T-cell infiltration and impaired T-cell quality due to the presence of an immunosuppressive tumor microenvironment, which affect the safety and limit efficacy of CD3-bispecific antibody therapy. In this review, we provide a brief status update of the CD3-bispecific antibody therapy field and identify intrinsic hurdles in solid cancers. Furthermore, we describe potential combinatorial approaches to overcome these challenges in order to generate selective and more effective responses.

## 1. Introduction

CD3-bispecific antibodies (CD3-BsAbs) are an emerging treatment modality in the field of cancer immunotherapy. BsAbs can recognize distinct antigens with each of their antigen-binding domains, in contrast to conventional Abs that recognize the same antigen with both Fab arms. The exception is IgG4, which has been reported to naturally exchange arms to attain bispecificity [1]. CD3-BsAbs act by simultaneous binding to a tumor-associated antigen (TAA) expressed on tumor cells and to CD3 on a T cell (CD3xTAA) [2]. Crosslinking of these two cell types by CD3-BsAbs allows the formation of an immunological synapse, similar to that of a natural T-cell receptor (TCR)/peptide–major histocompatibility complex (MHC) complex [3]. This synapse results in T-cell activation and thereby the secretion of inflammatory cytokines and cytolytic molecules that are able to kill the tumor cells in the process. The strength of CD3-BsAbs lies in the fact that any T cell could serve as an effector cell, regardless of TCR specificity, as for these BsAbs, TCR signaling does not require engagement of the antigen-binding domain of the TCR, but is initiated via CD3 [4]. Therefore, CD3-BsAbs can employ all available T cells and are not limited to tumor-specific T cells, contrary to the key requirement for effective immune checkpoint therapy [5].

CD3-BsAb therapy is a passive form of immunotherapy and shows striking kinship with the adoptive cell transfer of T cells expressing chimeric antigen receptor (CAR) transgenes [6]. CARs consist of TAA binding domains from antibodies directly linked to the intracellular CD3ζ chain and domains from costimulatory receptors (e.g., 4-1BB) and thereby activate T cells upon antigen recognition. CD3-BsAbs and CAR T cells are similar in many ways: both target a surface TAA, both exploit T-cell effector functions and both are successfully used in the clinic for hematological malignancies and show a similar type of toxicity profile [7,8]. Some disadvantages of currently clinically approved CAR T cells compared to CD3-BsAbs are: (1) patients are required to be lymphodepleted prior to infusion of CAR T cells, (2) CAR T cells have to be individually produced for each patient, whereas CD3-BsAbs can serve as off-the-shelf therapeutics, (3) CAR T cells remain in the patients after the tumor is cleared, resulting in continuous B-cell depletion in the case of CD19-targeting CAR T cells, whereas CD3-BsAbs are cleared from the blood over time and (4) unlike CD3-BsAbs, dosing cannot be adjusted to minimize adverse events [7,9]. Nevertheless, it will be important to learn from the CAR T cell field to potentially extrapolate new findings to the CD3-BsAb field.

Over the last few years, new insights in BsAb biology and enabling technologies resulted in the generation of many different formats of CD3-BsAbs, which was elaborately reviewed by Labrijn et al. [10]. As of December 2020, over 100 different CD3-BsAb formats are known, ranging from very small fragments containing two different variable domains without an Fc tail, conventional antibody structures (two Fab arms linked to an Fc tail) and larger structures with additional variable domains linked to the conventional antibody structure. These different formats determine important features, such as antibody half-life via neonatal Fc receptor (FcRn)-mediated recycling, immunogenicity, type of effector response via altered immune synapse formation and ability to penetrate in solid tumors [11]. The presence and functionality of the Fc tail determines whether the BsAb is able to bind to and activate Fc receptor (FcR)-expressing immune cells, which could lead to stronger inflammatory responses, but also allows activation of immune cells in the absence of TAA, potentially resulting in more severe adverse events (AEs) [12].

Currently, CD3-BsAbs show great potential for hematological cancers, with the FDA-approved blinatumomab (CD3xCD19) being successfully used in the clinic to treat some B-cell malignancies. Many other CD3-BsAbs are being tested in (pre)clinical studies for both hematological and solid tumors. However, contrary to the success of CD3-BsAbs in hematological malignancies, the effect of these antibodies in solid tumors is still rather limited [13]. This review will focus on essential hurdles for CD3-BsAbs for solid tumors, such as critical on-target off-tumor binding, sparse T-cell infiltration and quality of tumor-infiltrating lymphocyte (TIL) effector cells due to the presence of an immunosuppressive tumor microenvironment (TME). Lastly, we will discuss potential combination strategies to overcome these hurdles. 

## 2. Main Text

### 2.1. CD3-BsAbs in Hematological Malignancies

CD3-BsAbs received a lot of attention due to their success in hematological cancers. Blinatumomab (a CD3xCD19 BsAb without an Fc tail) was FDA approved in 2014 and is now successfully used in the clinic to treat patients suffering from relapsed or refractory B-cell precursor acute lymphoblastic leukemia (ALL) [14]. Over 40% of adult patients treated with blinatumomab show a complete or partial response and median overall survival is improved by several months compared to standard of care chemotherapy [15,16,17]. Unfortunately, most patients still relapse eventually after primary response to blinatumomab therapy. These relapses are currently being extensively investigated and the data have thus far indicated that relapses are frequently found at immune-privileged extramedullary locations and some relapses have lost CD19 antigen expression, but more research is required to further elucidate these resistance mechanisms [18,19]. 

Apart from blinatumomab, many other CD3-BsAbs are currently in clinical trials targeting well-established B-cell markers, like CD19, CD20, CD38 and B-cell maturation antigen (BCMA) and myeloid markers, like CD33 and CD123. For instance, in a phase I/II study, patients suffering from acute myeloid leukemia (AML) were treated with flotetuzumab (CD3xCD123 BsAb) and showed promising overall response rates (complete response with full, partial or incomplete recovery of blood cells) of 30% [20]. In another phase I/II study for patients suffering from diffuse large B-cell lymphoma (DLBCL), high-grade B-cell lymphoma (HGBCL) or follicular lymphoma (FL), epcoritamab (CD3xCD20 BsAb) therapy generated impressive responses: 44% complete response (CR) and 11% partial response (PR) for patients with DLBCL or HGBCL and 100% PR for patients with FL [21]. Comparable results were obtained with other CD3xCD20 bispecifics [22,23]. In NOD/SCID-gamma null (NSG) mice, REGN1979 (CD3xCD20) delayed tumor outgrowth better than rituximab, thereby further indicating the strength of CD3-BsAbs [24]. Interestingly, some of these trials target the same B-cell or myeloid antigens, however, with different CD3-BsAb formats. Therefore, these clinical studies could potentially inform on the role of different antibody formats’ treatment safety and efficacy. 

Clinical trials with blinatumomab revealed that cytokine release syndrome (CRS) is one of the major safety-related AEs [25]. The availability of CD19^+^ tumor cells and healthy B- and T cells in the same compartment allows acute and synchronic CD3-BsAb-mediated T-cell activation, followed by excessive release of inflammatory cytokines, such as IFN-γ, IL-6 and TNF-α, resulting in symptoms ranging from mild fever to multi-organ system failure [26]. However, CRS is not a specific problem for blinatumomab, but is observed for all CD3-BsAbs and CAR T-cell therapies in both hematological and solid cancer indications with CRS severities dependent on the type of therapy and target [27,28]. Preclinical research using a humanized mouse model showed that the primary mediator of CD3-BsAb-induced CRS was TNF-α produced by activated T cells, leading to massive secretion of inflammatory cytokines by monocytes [29]. The blockade of upstream TNF-α and downstream IL-1β or IL-6 can mitigate CRS [29,30,31]. Others have reported that step-up dosing, or subcutaneous administration of CD3-BsAbs, decreased the extent of CRS [32,33]. Furthermore, several preclinical studies in mouse and cynomolgus monkey models showed that reducing CD3 affinity could reduce treatment-induced cytokine levels [34,35,36,37].

### 2.2. Historical Perspective and Current Status of CD3-BsAbs in Solid Cancers

Despite the fact that CD3-BsAbs are mostly known for their use in hematological malignancies, the first European medicines agency (EMA)-approved CD3 bispecific antibody was catumaxomab, a CD3xEpCAM BsAb for the intraperitoneal treatment of epithelial cell adhesion molecule (EpCAM)-positive malignant ascites [38]. This antibody was actually trifunctional, as its Fc was able to bind FcR-expressing cells and induced strong immunological responses [39]. Severe liver toxicity was also observed due to the activation of Kupffer cells when administered intravenously [12]. Catumaxomab was eventually withdrawn for commercial reasons in 2017, but taught the field an important lesson about the potential dangers of the presence of an active Fc in CD3-BsAbs. All current full length CD3-BsAbs in development contain Fc-silent backbones with mutations impairing the binding of FcγR and C1q [10]. Moreover, preclinical studies showed that Fc-silenced full length CD3-BsAbs improved T-cell trafficking towards the tumor and induced better anti-tumor responses. Wang et al. showed that CD3-BsAbs with an active Fc backbone failed to drive T cells to the tumor, but instead induced either T-cell depletion or the accumulation of T cells in the lungs [40]. This observed effect was attributed to the capacity of the Fc backbone to be bound by FcγR-expressing myeloid cells. Fc-silenced CD3-BsAbs did not lead to sequestration of T cells in the lungs, but they arrived in the tumor. More importantly, therapeutic efficacy was greatly improved in Fc-silenced CD3-BsAb-treated mice. A similar trend was also observed in a syngeneic mouse model, where CD3xTrp1 (tyrosinase-related protein 1) was used to treat Trp1-positive B16F10 tumor cells [41]. 

As of December 2020, no CD3-BsAbs are approved for the treatment of solid tumors in the clinic. However, many different targets are being explored in clinical studies, of which most are focusing on classical TAAs, such as carcinoembryonic antigen (CEA), epidermal growth factor receptor (EGFR), EpCAM, HER2 and prostate-specific membrane antigen (PSMA). Other TAAs are also being explored (see Table 1 for an elaborate list). Most of these studies simply inject CD3-BsAbs, however, in some studies, these Abs piggyback with infused T cells as “bispecific-armed T cells”. Furthermore, this table also includes CD3-BsAb formats based on affinity-enhanced TCR-like domains that recognize peptide–human leukocyte antigen (HLA) complexes (immune mobilizing monoclonal T-cell receptors against cancer (ImmTACs)) [42]. Multiple other TAAs are currently pursued in preclinical studies hoping to make their way to the clinic, including B7-H4, CD133, CD155, claudin 6 (CLDN6), cellular mesenchymal to epithelial transcription factor (C-MET), ephrin receptor A10 (EphA10), folate receptor 1 (FOLR1), HLA-A*24:survivin 2B_80-88_, integrin β4 (ITGB4), P-cadherin, prolactin receptor (PRLR), receptor tyrosine kinase-like orphan receptor 1 (ROR1), TNF-related apoptosis-inducing ligand receptor (TRAIL-R2), transferrin receptor (TfR) and tumor-associated calcium signal transducer 2 (Trop-2) [43,44,45,46,47,48,49,50,51,52,53,54,55,56,57,58,59,60,61,62,63,64,65,66,67,68,69].

Most of these studies are currently still enrolling patients and we are only starting to get a view on CD3-BsAb therapy safety and efficacy in solid cancers. First, in three different studies, patients were treated with an i.p. infusion of catumaxomab, which resulted in frequent but manageable toxicities and increased time between paracentesis in all studies and even a significant improvement in overall survival (OS) in one study [38,70,71]. Other clinical trials in solid tumors reported dose-limiting toxicities (DLTs) for CD3-BsAbs targeting CEA, EpCAM and HLA-A*02:01:gp100 [70,71,72,73]. These toxicities consisted of abnormal liver parameters, colitis, CRS, diarrhea, dyspnea, hypotension, hypoxia, respiratory failure and tachycardia. Some of these toxicities were caused by tumor lesion inflammation, however, most were reversible upon treatment discontinuation. Responses to CD3-BsAbs varied from only 1.5% partial response (PR) [70], up to 15% PR and 46% stable disease (SD) [73] and everything in between [73,74,75]. Pasotuxizumab, a CD3-BsAb targeting PSMA, obtained the most impressive results with two long-term responders, of which one had marked regression of soft tissue and bone metastases [74]. Overall, some evidence for efficacy induced by CD3-BsAbs in solid tumors has been found, however, with only a handful of long-term survivors, some partial responses and the occurrence of multiple DLTs, the development of CD3-BsAbs in solid tumors lags behind that in hematological malignancies.

### 2.3. Hurdles in Solid Tumors

The observation that CD3-BsAbs seem more efficacious in hematological malignancies than in solid tumors can be attributed to several challenges that are specific to solid tumors. The first hurdle is on-target off-tumor toxicities, as these seem less forgiving for TAAs selected for solid tumor targeting, when compared to hematologic TAAs [76]. In the case of hematological cancers, the temporary depletion of B cells or myeloid subsets is reversible, as long as hematopoietic stem cells are not targeted, allowing replenishment of the blood pool. However, solid tumor TAAs are often also expressed on tissues of healthy organs, which can lead to immune pathology and organ failure with potential fatality, as shown in a preclinical mouse study using a CD3-BsAb targeting EGFR [75]. Critical selection of a tumor-specific TAA is thus crucial.

The second hurdle is the availability of effector cells in the TME. For hematological malignancies, cancer cells in the blood are surrounded by T cells, allowing the CD3-BsAb to draw from an endless pool of effector cells, whereas solid tumors require T-cell infiltration for therapeutic efficacy. In this context, three immune landscapes have been described: (1) “inflamed” tumors, which are infiltrated by immune cells and frequently respond to immune checkpoint therapy [77], (2) “immune desert” tumors, which have a reduced or absent immunogenicity, resulting in very few primed tumor-specific T cells that home to the tumor [78] and (3) “immune-excluded” tumors, which display T-cell infiltration in the stroma, but not the tumor nests [79,80]. For these immune-excluded tumors, the deposition of extracellular matrix (ECM) components in the stroma results in a physical barrier surrounding the tumor parenchyma. Apart from this physical barrier, the secretion of soluble factors such as transforming growth factor beta (TGF-β) and C-X-C motif chemokine ligand 12 (CXCL12) further frustrates T-cell infiltration. Since T-cell trafficking to solid tumors can be scarce in immune desert or immune-excluded tumors, CD3-BsAbs might have only a few T cells available in the TME (see Scheme 1 for T-cell development and trafficking). The requirement of T-cell infiltrate for effective CD3-BsAb therapy was described by Ströhlein et al. in a clinical study with catumaxomab [81]. 

The third and final hurdle concerns the quality of infiltrating T cells. TILs can be dysfunctional, with impaired ability to proliferate and produce cytolytic molecules, including granzymes and perforins [89]. Immunosuppressive cells in the TME, including cancer-associated fibroblasts (CAFs), myeloid-derived suppressor cells (MDSCs) and regulatory T cells (T_regs_) produce factors such as TGF-β, IL-10, indoleamine 2,3-dioxygenase (IDO) and arginase, which hamper T-cell metabolism and activation [90]. Additionally, effector T cells were shown to exhibit an “exhausted” profile due to chronic antigen stimulation, as witnessed by the expression of inhibitory immune checkpoints, such as programmed cell death protein 1 (PD-1) and cytotoxic T-lymphocyte antigen 4 (CTLA-4) [91]. Furthermore, it has been reported that CD3-BsAbs might induce TIL apoptosis via activation-induced cell death, which hampers a strong anti-tumor response [48]. Some of the discovered TME obstacles have only been described in resistance upon immune checkpoint therapy, however, we expect these hurdles to also play a role in CD3-BsAb therapy. An overview of these hurdles is shown in Figure 1.

### 2.4. Solutions and Opportunities

#### 2.4.1. Mitigating of On-Target Off-Tumor Toxicities

Most alterations in cells during the process of oncogenesis affect intracellular circuits involved in the cell cycle, survival and invasive growth [92]. As such, the surface proteome is relatively conserved between cancer cells and their healthy counterparts, with the exception of tyrosine kinase growth receptors, e.g., EGFR family members, which are overexpressed and sometimes truncated in extracellular domains [93]. The search for suitable TAAs for targeting by CD3-BsAbs is therefore complicated, as these targets should be surface proteins that are exclusively expressed by tumor cells and absent in healthy cells. HLA molecules can present small neo-antigenic peptides derived from mutated proteins or peptides from tumor virus proteins and these peptide/HLA complexes can serve as highly cancer-specific TAAs [94]. Their disadvantage is that most identified neoantigens are patient specific and viral antigens are observed in only a subset of cancers [95,96]. A lot of research is currently being performed to identify new and more common neoantigens, which may result in promising future TAAs for CD3-BsAb therapy [97]. The next best targets are overexpressed proteins on tumor cells compared to healthy cells, which is the case for most of the clinically targeted TAAs, such as CEA, EGFR, EpCAM and HER2 [98,99]. Targeting these TAAs offers some selectivity of tumor cells over healthy cells dependent on the extent of overexpression, however, healthy tissues can still be affected. 

Some of the most differentially expressed genes in cancer are actually intracellular proteins, which cannot be reached by conventional antibodies [100]. This intracellular proteome is only approachable via HLA class I molecules, which present them to the outer world. These surface peptide/HLA complexes can be targeted with peptide-specific antibody formats or by TCR molecules, such as ImmTACs or T-cell engaging receptor (TCER) molecules [101,102]. The TCR arms of ImmTACs and TCERs are affinity enhanced to low nM ranges to obtain sufficient TAA binding strength to the cancer cell in order to successfully engage effector T cells with the CD3-binding arm. However, major disadvantages of TCR-like CD3-BsAbs are HLA restrictions and vulnerability towards HLA downregulation in the tumor. ESK1, an ImmTAC recognizing intracellular Wilms’ tumor 1 (WT1) antigen presented in HLA-A*02:01, was able to lyse WT1-positive tumor cells in vitro and reduce tumor outgrowth of AML, ALL and mesothelioma tumors in NSG mice [103]. A clinical trial with the ImmTAC tebentafusp targeting gp100 presented on HLA-A2*02:01 has successfully been completed and response rates of 15% PR and 46% SD were reported. Unfortunately, DLTs in the form of hypotension were observed in four patients receiving the highest dose [73]. These intracellular targets are both overexpressed TAAs and still seem to generate on-target off-tumor toxicity (as described for tebentafusp). The development of ImmTACs targeting more specific TAAs, such as neoantigens derived from highly conserved Ras mutations in various cancers, E6 and E7 peptides from human papilloma virus (HPV)-induced cancers or T-cell epitopes associated with impaired peptide processing (TEIPP) antigens for cancers with defects in transport associated with antigen processing (TAP) function, could be promising [104,105,106].

To improve tumor selectivity and specificity and mitigate on-target off-tumor toxicities, the TAA avidity could be increased, for instance, by generating so-called 2:1 CD3-BsAbs. These 2:1 CD3-BsAbs contain a second TAA binding fragment, resulting in a CD3xTAAxTAA bispecific antibody [107]. Slaga et al. showed that specificity for high-expressing HER2 cells was significantly increased when using a 2:1 HER2-targeting CD3-BsAb in vitro [108]. More importantly, tumor growth of high HER2-expressing tumors in NSG mice was efficiently delayed by the 2:1 CD3-BsAb, whereas no anti-tumor efficacy was observed in low HER2-expressing tumors. In contrast, the 1:1 CD3-BsAb was able to effectively delay tumor growth in both high and low HER2-expressing tumors (used here as a model for healthy tissue). In cynomolgus monkeys, i.v. infusion of the 2:1 CD3-BsAB did not result in an increase in C-reactive protein (CRP), T-cell activation or alanine or aspartate aminotransferase (ALT and AST) levels in blood upon exposure to the endogenous expression of HER2 on healthy cells. However, a direct comparison between these two formats was not feasible, as similar cynomolgus monkey data were not generated for the 1:1 CD3-BsAb. A similar improved selectivity was observed for a 2:1 CEA-targeting CD3-BsAb [109]. This CD3-BsAb was tested in patients with advanced CEA-positive carcinomas and displayed signs of anti-tumor effects (5% PR, 11% SD) with a manageable toxicity profile, which was most likely associated with tumor lesion inflammation [72]. In this 2:1 format, TAA affinity plays a very important role: when the TAA affinity is too high, there is no increased specificity, as the BsAb can still bind to low levels of the TAA expressed on healthy cells. On the other hand, if the TAA affinity is too low, the potency of the CD3-BsAb will be compromised. Therefore, modulating TAA affinity could be seen as a tight balance between specificity and efficacy [110]. In a similar fashion, specificity could also be improved for 1:1 CD3-BsAbs by lowering TAA affinity, however, due to the lower avidity compared to 2:1 BsAbs, it is expected to be harder to achieve the optimal balance [111].

A conceptual novelty in this area is the generation of CD3-BsAbs as prodrugs that are activated in the TME. Differences in physiological features, such as hypoxia-related low pH, excessive production of ECM and increased proteolysis, distinguish solid tumors from healthy tissues [112,113]. These differences warranted the development of CD3-BsAbs with binding regions that are masked with protease-cleavable linkers. Boustany et al. developed a masked CD3xEGFR BsAb, which blocked both CD3 and EGFR binding [114]. The binding of this BsAb to CD3 and EGFR was strongly reduced in vitro in the absence of proteases, whereas in vivo anti-tumor efficacy was retained. In cynomolgus monkeys, the maximum tolerated dose (MTD) for masked CD3xEGFR was 60-fold higher than the unmasked variant. Additionally, the masked variant greatly prolonged plasma concentrations at higher dosing concentrations. Very recently, Panchal et al. described the development of conditional bispecific-redirected activation (COBRA) T-cell engagers [115]. This format separates the α-CD3 V_H_ and V_L_ via a matrix metallopeptidase 9 (MMP9)-degradable linker, thereby only allowing CD3 binding after linker cleavage, while constantly allowing EGFR and serum albumin binding (to increase half-life). In co-cultures of T cells with tumor cell lines, a dependency on the presence of MMP9 was demonstrated for T-cell-mediated cytotoxicity. In vivo, COBRA BsAb could completely eradicate established HT-29 colorectal tumors in NSG mice, whereas their non-cleavable BsAb counterpart displayed no anti-tumor activity. Furthermore, Geiger et al. described a folate receptor 1 (FOLR1)-targeting CD3-BsAb (Prot-FOLR1-TCB) that linked a protease-cleavable anti-idiotypic anti-CD3 mask to the CD3 arm [116]. This masked the CD3-binding domain of the BsAb and was cleaved by proteases produced in the TME, resulting in selective T-cell activation after the addition of protease in vitro. In humanized mice, Prot-FOLR1-TCB was able to delay tumor outgrowth to a similar extent to the non-masked BsAb. Other approaches that have split the anti-CD3 modality into two separate components, which only functionally recombine at the tumor surface when both bind to separate TAAs, are also being explored [117,118].

Another way to mitigate on-target off-tumor toxicities is to alter CD3-BsAb distribution, for example, by the modification of CD3 affinity. In a preclinical mouse study, the distribution of radiolabeled CD3xHER2 BsAbs with different CD3 affinities was followed by single-photon emission computed tomographic (SPECT) imaging. This study showed that CD3-BsAbs with a high affinity CD3 arm accumulated in T-cell-rich tissues, such as the spleen and lymph nodes, whereas CD3-low affinity BsAbs accumulated mainly in the HER2^+^ tumor, thereby affecting the biodistribution and treatment outcome [119]. Instead of the systemic administration of (conditionally active) CD3-BsAb, another option to alter distribution could be to administer CD3-BsAbs intratumorally. Although this would not completely prevent systemic spreading, as some CD3-BsAb will probably enter the blood by diffusion, local administration can strongly reduce on-target off-tumor toxicity [120]. Although local administration is possible under ultrasound guidance for non-superficial tumors, this method is still complicated because multiple injections are required. Alternatively, delivery systems can be exploited that would selectively release or produce CD3-BsAb in the tumor. One such method is the use of transduced (tumor-specific) T cells, that are engineered to express CD3-BsAbs upon T-cell activation, also called the secretion of T-bsAbs by engineered (STAb)-T cells [121]. Iwahori et al. generated STAb-T cells recognizing the erythropoietin-producing hepatocellular carcinoma A2 (EphA2) antigen that produced CD3xEphA2 BsAb upon T-cell activation [122]. They showed effective anti-tumor activity in U373 glioma and A549 lung tumors in NSG mice, while systemic exposure to the CD3-BsAb seemed to be minimal, as indicated by the absence of human cytokines in peripheral blood. Alternatively, BsAb constructs can be expressed in producer lines or non-specific T cells, however, this approach is not well developed at the moment [123,124]. Oncolytic virus (OV) was also used as a delivery vehicle as it selectively replicates in transformed cancerous cells over healthy cells [125]. Fajardo et al. described an oncolytic adenovirus encoding CD3xEGFR BsAb and observed a modest but significant delay in tumor outgrowth when used either by intratumoral or i.v. administration in NSG mice [126]. In a different study, oncolytic measles virus encoding CD3xCEA BsAb was developed and used to treat patient-derived colorectal cancer xenografts in NSG mice [127]. Tumor outgrowth was moderately delayed in mice treated intratumorally, without detectable BsAb in serum. However, when mice were treated i.v., only low BsAb levels were detected in the tumor in contrast to high BsAb concentrations in peripheral blood. Other groups also used OVs encoding CD3-BsAbs targeting EpCAM, Eph-A2 and CD44v6 and observed anti-tumor activities against several tumor models [128,129,130]. However, in all of these studies, OV encoding CD3-BsAb was administered locally and toxicity evaluation was not reported. An overview of mitigation strategies to overcome off-tumor on-target toxicities is depicted in Figure 2.

#### 2.4.2. Increasing the Number of Intratumoral T cells

Tumors can be classified in three categories regarding T-cell infiltration: immune desert, immune-excluded and inflamed [79]. Immune desert tumors barely contain T cells at all, not in the tumor nests nor surrounding the rims, thereby potentially limiting CD3-BsAb therapy efficacy. Interestingly, intratumoral OV administration can ignite T-cell influx in immune desert tumors. The replication of oncolytic virus can generate an interferon response in the TME and induce an innate and adaptive antiviral immune response [131,132,133]. We used this concept to pre-treat immune desert murine tumors (B16F10 melanoma and (LSL-Kras^G12D^, LSL-Trp53^R172H^, Pdx-1-Cre) KPC pancreatic carcinoma) with OV, which induced sensitization and generated major T-cell influx peaking around day 7, allowing strong tumor regression upon CD3-BsAb treatment [134]. In the absence of OV sensitization, CD3-BsAb did not even delay tumor growth, underlining the importance of an inflammatory TME. Of note, we found that the simultaneous administration of CD3-BsAb and OV did not provide survival benefit, indicating that the timing of OV and CD3-BsAb is an important aspect.

Immune-excluded tumors have T cells surrounding the tumor, but penetration into tumor beds is hindered by physical barriers or soluble factors. Efforts to improve therapy efficacy in these types of tumors should therefore focus on removing these obstructions. The physical barrier is mainly formed by ECM structures, which forces T cells to move along areas of increased stiffness instead of following the chemokine gradient in a process called haptotaxis [135]. This barrier consists of proteoglycans and fibrous proteins such as collagen, elastin and laminin, which are mainly produced by CAFs, but also by tumor cells and stellate cells [135,136]. The ECM could be targeted by the direct destruction of ECM components, such as collagen and hyaluronic acid (HA), a process which is also studied in the context of chemotherapeutic drug delivery to the tumor [137,138,139]. Guan et al. used hyaluronidase to break down HA, which increased the infiltration of tumor-specific T cells and greatly improved treatment efficacy in a B16.OVA melanoma mouse model [140]. Not only ECM components, but also their cellular producers could be targeted. CAFs are the major producers of ECM products and highly express fibroblast activation protein (FAP), which constitutes an attractive target for immunotherapy. In several mouse tumor models, T-cell infiltration was increased upon CAF targeting using DNA vaccines or fibrosis inhibitors [141,142,143]. OV encoding a CD3-BsAb targeting FAP was elegantly used in several studies to kill CAFs and simultaneously enhance T-cell infiltration [144,145,146]. 

Apart from creating a physical barrier, CAFs can also influence T-cell infiltration by various secreted molecules [147]. CAFs are the major source of the chemokine CXCL12, which has been implicated to mediate T-cell exclusion in solid tumors [148]. The inhibition of CXCL12 or its receptor CXCR4 resulted in increased T-cell infiltration and rendered tumors vulnerable towards checkpoint inhibition therapy in mouse models for pancreatic and colorectal cancer [148,149]. Furthermore, TGF-β has been implicated in hampering T-cell infiltration. This immunosuppressive cytokine is produced by CAFs, but also many other cells, including T_regs_ and M2 macrophages [147]. Similar to CXCL12 signaling inhibition, blocking TGF-β signaling resulted in more T-cell infiltration and increased sensitivity to checkpoint inhibition therapy in multiple mouse models for breast cancer and colorectal cancer [150,151,152]. Post-translational modifications of secreted factors in the TME can also affect T-cell attraction, as was reported for CCL2 [153]. Nitrification of CCL2 by reactive nitrogen species (RNS) in the TME resulted in T cells being stuck in the stroma surrounding the tumor cells. The inhibition of RNS production greatly improved T-cell infiltration in several mouse tumor models and thereby improved survival as a monotherapy or in combination with adoptive cell transfer. An overview of solutions to overcome sparse T-cell infiltration is shown in Figure 3.

#### 2.4.3. Improving the Quality of T-cell Responses

When the CD3-BsAb and sufficient T cells have finally reached the tumor, they are faced with another challenge: a hostile and immunosuppressive TME. Firstly, immune checkpoint ligands are known to be expressed in the TME, such as programmed death ligand 1 (PD-L1) and HLA-E [154,155]. PD-L1 is upregulated on tumor, stromal and immune cells upon local interferon release by immune cells [156,157,158]. Most intratumoral T cells already express PD-1 and the CD3-BsAb-mediated activation of T cells further stimulates the expression of this inhibitory co-receptor, thereby hampering effector functions and treatment efficacy [159,160]. Combination therapies of CD3-BsAbs and checkpoint blockade have been widely investigated in multiple in vitro studies and mouse models and resulted in improved tumor control [44,161,162]. Interestingly, Osada et al. reported that PD-1/PD-L1 blockade could not improve T-cell functioning in an in vitro setting if blockade is applied after T cells have engaged tumor cells [163]. Exhausted cells could no longer be rescued when blockade was applied too late, thereby emphasizing the importance of timing. The combination of CD3-BsAbs with checkpoint blockade is also being explored in several clinical studies (NCT03319940, NCT03531632, NCT03406858, NCT03272334, NCT03564340, NCT03792841, NCT04590781 and NCT02324257). The NCT02324257 study (CD3xCEA in combination with atezolizumab (anti-PD-L1) for patients with advanced CEA-positive tumors) has been completed and seemed to be in line with previous preclinical results: the combination of CD3-BsAbs with checkpoint blockade showed better anti-tumor responses when compared to CD3-BsAb monotherapy and they found no evidence of increased toxicities [72]. Therefore, this combination holds great promise.

Unfortunately, not only immune checkpoints have the capability to dampen T-cell function. Due to their rapid glycolysis-dependent proliferation, tumor cells generate a hypoxic and low-glucose TME [164]. In these hypoxic conditions, hypoxia-induced factors (HIFs) initiate the expression of CD39 and CD73 by multiple cell types in the TME; CD39 and CD73 convert free ATP in the TME into adenosine [165]. Adenosine has been reported to counteract TCR activation by binding to adenosine A_2A_ receptors via protein kinase A and cyclic Amp signaling, which suppresses the effector functions of T cells [161,162]. Furthermore, low glucose concentrations have been reported to dampen the anti-tumoral cytokine production and survival of effector T cells, as they rely heavily on glucose for their functioning [166,167]. Apart from these tumor-mediated factors, stromal cells, infiltrating T_regs_ and MDSCs secrete other immunosuppressive factors, such as IDO, TGF-β, IL-10 and arginase. IDO has been reported to degrade the amino acid tryptophan into kynurenine, resulting in decreased effector function via downregulation of the CD3 ζ-chain, and induce apoptosis in T cells [164,168]. TGF-β is able to suppress T-cell effector function and inhibit the differentiation of CD4^+^ cells into effector cells, while promoting T_reg_ differentiation [169,170,171]. IL-10 has been described to induce T-cell anergy and prevent the development of new effector T cells by decreasing antigen presentation and costimulation on antigen-presenting cells (APCs) [172,173]. Arginase breaks down the amino acid arginine and, similar to IDO, results in the downregulation of the CD3 ζ-chain and thereby decreased cytotoxic and proliferative capacity of T cells [174,175]. 

Most of these immunosuppressive factors do not only act on T cells but also on other cells in the TME to generate a negative feedback loop and dampen the inflammatory response. The inhibitory effect on T cells could be overcome by decreasing the amount of immunosuppressive signals, which could be accomplished by blocking receptor–ligand interactions for these molecules. Several studies have been published reporting improved T-cell effector function after blocking adenosine, arginase, IDO, IL-10 and TGF-β [176,177,178,179,180,181]. CD3-BsAb therapy has thus far only been combined preclinically with IDO blockade. Hong et al. showed improved in vitro killing and in vivo tumor control of EpCAM- and IDO-positive murine breast cancer cells using a combination of an EpCAMxCD3 BsAb with IDO blockade [182]. The cells producing these immunosuppressive factors can also be targeted, which is already being extensively studied for CAFs, as described above. MDSCs are popular targets as well, with therapies being developed to deplete them and prevent migration into the TME, resulting in improved anti-tumor activity in combination with several different types of immunotherapy [183,184,185,186,187,188,189,190]. Currently, no combinations of CD3-BsAbs and MDSC targeting have been reported for solid tumors. However, several studies in hematological malignancies showed that CD3xCD33 BsAbs mediated both AML and MDSC killing, yielding promising treatment outcomes [191,192]. Finally, depleting T_regs_ in combination with CD3-BsAb treatment could be favorable in two ways: (1) T_regs_ secrete immunosuppressive cytokines such as IL-4, IL-10 and TGF-β and, more importantly, (2) T_regs_ are suggested to be activated by CD3-BsAbs, resulting in a dampened treatment effect [193,194]. Forkhead box protein P3 (FoxP3)-positive T_regs_ highly express OX40, CTLA-4 and CD25 and can be depleted to achieve stronger anti-tumor effects in combinatorial strategies [195,196,197,198,199]. One study investigated the effect of combining CD3xEGFR-armed T cells with T_reg_-depleting ipilimumab (anti-CTLA-4) on T-cell activation and proliferation when co-cultured with tumor cell lines or primary tumor cells and found enhanced T-cell-mediated cytotoxicity and increased T-cell proliferation [200]. Thus far, there are some promising preclinical results for combining CD3-BsAbs with IDO blockade, MDSC depletion and T_reg_ depletion, warranting further exploitation of these combinations.

Instead of decreasing T-cell inhibitory signals, another approach would be to trigger stimulatory receptors on T cells, which could be induced by administering agonistic antibodies for these receptors on T cells, such as CD28 and 4-1BB. This approach parallels the addition of a costimulatory intracellular signaling domain to improve efficacy for second generation CAR T cells [201]. The combination of CD3-BsAbs with costimulatory antibodies has been successfully used in many different tumor models in mice [202,203]. Chiu et al. showed in a humanized mouse model that combination of a CD3xPSMA BsAb with a costimulatory agonistic 4-1BB Ab greatly enhanced anti-tumor efficacy [204]. The combination successfully improved the survival of mice bearing large tumors in contrast to CD3-BsAb monotherapy and, more importantly, generated a memory response that protected surviving mice from a second tumor challenge. However, weight loss was reported in the mice receiving the combination treatment, which is in line with reported toxicities for the administration of bivalent agonistic 4-1BB costimulatory Abs [205,206]. Conditional costimulation only in the tumor TME can be generated by CD28xTAA BsAbs [207]. Using this localized costimulation, Skokos et al. observed no toxicities in in vitro assays as well as in cynomolgus monkey toxicity studies, while these combinations still displayed impressive enhancements in anti-tumor activity in various mouse models [207]. Therefore, the combination of CD3-BsAbs with TME-targeted costimulatory BsAbs seems promising. Furthermore, additional costimulation has been reported to protect T cells from Fas-mediated apoptosis after activation by CD3-BsAb [208]. Currently, a clinical trial is investigating the combination of CD3xMUC1 with CD28xMUC1 and we are looking forward to seeing if the promising preclinical results will translate to clinical efficacy (NCT04590326). Finally, T-cell-sustaining cytokines can be coinjected, or engineered onto CD3-BsAbs. Rossi et al. reported that IFN-α enhanced T-cell activation and delayed tumor outgrowth in two mouse models [209]. Schmol et al. linked IL-15 to a bispecific natural killer (NK) cell engager and showed enhanced NK cell proliferation, activation and survival in vitro. This finding could potentially be translated to CD3-BsAbs as well, since IL-15 also promotes T-cell survival [210]. An overview of the solutions to improve T-cell quality is depicted in Figure 4.

## 3. Future Perspectives

CD3-BsAbs are an emerging and promising class of immunotherapy due to their impressive treatment outcomes in hematological malignancies, however, prominent anti-tumor efficacy in solid tumors still needs to be delivered clinically. Particularly, a whole array of hurdles arises in solid cancers, ranging from on-target off-tumor toxicities and the absence of T-cell infiltration in the TME, to hampered T-cell function attributable to a hostile and immunosuppressive microenvironment. Due to continuous research efforts, more tumor-specific TAAs become available every year for use in CD3-BsAb formats. Combined with constantly evolving technologies allowing conditional masking of BsAbs, on-target off-tumor effects should be manageable in the near future. Some interesting pre-clinical concepts have been published to enhance T-cell infiltration in the tumor, such as pre-treatment with OV to facilitate massive T-cell infiltration and create an inflammatory TME. OV treatment seems most promising, as the inflamed TME also contributes to the quality of the TILs. Many options are available in terms of improving TIL quality, however, apart from checkpoint blockade or costimulation, only very few of them have been tested in combination with CD3-BsAbs. Nevertheless, based on elegant preclinical studies, we are convinced that a combination CD3-BsAbs with (tumor-targeted) costimulation is able to overcome many of the hurdles set by the TME of solid tumors.

We anticipate a future where the immune landscape of the tumor from a biopsy guides the selection of the best treatment combination. However, since many of these combinations have only just started to emerge, it will be intriguing to follow the results of new pre-clinical studies and see how those results translate to the clinic. Ultimately, based on these novel approaches, we foresee a bright future for CD3-BsAb-based therapy in solid tumors and are interested to see if comparable anti-tumor efficacy can be observed in solid cancer as seen in hematological cancers.

## Data Availability

No new data were created or analyzed in this study. Data sharing is not applicable to this article.

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
