# Peer review of "Overcoming Challenges for CD3-Bispecific Antibody Therapy in Solid Tumors"

_cancers, 2021, doi:10.3390/cancers13020287_

Round 1

Reviewer 1 Report

This is an interesting, comprehensive, well-written and structured review that gathers the latest updates on CD3-bispecific antibody therapy field and the challenges related to their use in solid tumors.

Minor comments

  1. Page 8, Lines 198-211: This paragraph is neither consistent with the content of the section, nor does it provide any new information, so I suggest removing it.
  2. Page 2: If pre-clinical or clinical studies are available, it would be interesting the authors to mention any comparative data on the use of CD3xCD20 bispecifics vs rituximab in hematological malignancies.

Author Response

Dear Reviewer,

Thank you for the kind words and review of our manuscript. We have made some small changes according to your feedback:

  1. Page 8, lines 209-223: This paragraph was intended as a box about T cell development and trafficking, describing some background regarding the T cell infiltration hurdle. We now clearly put a frame around the text to improve clarity and separate it from the main text.
  2. Page 3, lines 103-105: We found a preclinical head-to-head study comparing a CD3xCD20 bispecific antibody (REGN1979) with rituximab and added the following sentence to the manuscript: “In NSG mice, REGN1979 (CD3xCD20) delayed tumor outgrowth better than rituximab, thereby further indicating the strength of CD3-BsAbs” and added reference 31.

We hope these changes are satisfactory.

Best regards,

Jim Middelburg

Reviewer 2 Report

Middelburg and colleagues presented a review on CD3 bispecific antibodies that are in development in solid tumors. They reviewed the current literature and elucidated up-to-date development programs together with the biological backbone and mechanisms of the drugs.

Minor comments:

  • blinatumomab is approved only for ALL, not for various hematological malignancies
  • page 2 line 45-50: most of the limitations listed for CAR-t cells are to be overcome (e.g., with the production of off the shelf allogeneic CAR-t or with the incorporation in the viral vector of suicide genes). Without enrich this section, I suggest referring to the limitations you listed as limitations of commercially available CAR-T
  • page 3 line 05 and on CRS is a major issue of CAR-T, and is reported also linked to BsAbs; however the magnitude of the effect is largely different for every drug; CD19 commercially available carT har approx 50% incidence of serious CRS, blinatumomab less than 2%. Other bispecific (e.g. flotetuzumab) may have a CRS incidence more similar to CAR-T than to blina. Differences in risk for each drug has to be taken into account, and it is impossible to put in a box CAR-t together with blina and other BsAbs.
  • Furthermore, translating tumor microenvironment concepts, that were largely demonstrated to be important in patients who received immune checkpoint-targeting agents, may partially be misleading with BsAbs. I suggest a more cautious wording.

Author Response

Dear Reviewer,

Thank you for reviewing our manuscript and your insightful comments. We have made some changes according to your feedback:

  • Page 2 line 74: We added the word “some” to the sentence to indicate that blinatumomab is not approved for all B cell malignancies, but only particular malignancies and then mention in line 85 that it is approved for patients suffering from ALL.
  • Page 2 lines 52-53: We added the words “currently clinically approved” CAR T cells to indicate that the stated problems are only applicable to these CAR T cells as you suggested. We hesitate to include more text as our review is not focused on CAR T cells.
  • Page 3 line 114: We added the words “with CRS severities dependent on the type of therapy and target” to the sentence describing that CRS is encountered both for CD3-BsAb therapies as in CAR T cell therapy and added reference 35. We think this should clarify the issue of CRS in the two therapeutic strategies.
  • Page 9, lines 234-236: To the section describing the hurdle in the TME, we added the sentence “Some of the discovered TME obstacles have only been described in resistance upon immune checkpoint therapy, however we expect these hurdles also to play a role for CD3-BsAb therapy.” We think this should indicate that not all of them have been found for CD3-BsAbs and then in the solution section, people can read for each small component whether this has been done for CD3-BsAbs.

We hope these changes are satisfactory.

Best regards,

Jim Middelburg

Reviewer 3 Report

Therapy with CD3-bispecific antibodies is a promising therapeutic principle in hematologic neoplasms. In contrast to current cellular therapies, antibodies have many advantages. However, there are important limitations for solid tumors, which the authors clearly present. Table 1 presents numerous clinical trials that are now investigating whether new treatment options are emerging after all. Problems with the trials arise primarily from undesirable side effects and lack of therapeutic response. In the following, the authors present the three main challenges for the clinical use of CD3-bispecific antibodies. This is followed by the essential part of the paper, a presentation of strategies to overcome these problems.
The work is very well structured, builds on current literature and is informatively illustrated. The labeling of the 2nd chapter (Main Text) is unfortunate. The box "T-cell development and trafficking" should not be chapter 2.4 and should actually be framed.

Author Response

Dear Reviewer,

Thank you for the kind words and review of our manuscript. We have made some small changes according to your feedback:

Page 8, lines 209-223: We have removed the words “chapter 2.4” and put a frame around the text of the box.

We hope these changes are satisfactory.

Best regards,

Jim Middelburg